# The Use of the Bolk Model for Positive Health and Living Environment in the Development of an Integrated Health Promotion Approach: A Case Study in a Socioeconomically Deprived Neighborhood in The Netherlands

**DOI:** 10.3390/ijerph19042478

**Published:** 2022-02-21

**Authors:** Herman A. van Wietmarschen, Sjef Staps, Judith Meijer, J. Francisca Flinterman, Miek C. Jong

**Affiliations:** 1Department Nutrition & Health, Louis Bolk Institute, Kosterijland 3-5, 3981 AJ Bunnik, The Netherlands; s.staps@louisbolk.nl; 2Saba Cares, Paris Hill Road 12, The Bottom, Saba, The Netherlands; judith.meijer@phd.uvh.nl; 3Humanism and Social Resilience, University of Humanistic Studies, Kromme Nieuwegracht 29, 3512 HD Utrecht, The Netherlands; 4Healthy Living Department, Health Promotion and Health Innovation, GGD Amsterdam, Nieuwe Achtergracht 100, 1018 WT Amsterdam, The Netherlands; fflinterman@ggd.amsterdam.nl; 5Department of Health Sciences, Mid Sweden University, Holmgatan 10, 851 70 Sundsvall, Sweden; miek.jong@uit.no; 6National Research Center in Complementary and Alternative Medicine (NAFKAM), Department of Community Medicine, UiT the Arctic University of Norway, 9037 Tromsø, Norway

**Keywords:** public health, living environment, positive health, health promotion, integrated health

## Abstract

Background. Despite considerable efforts, health disparities between people with high and low socioeconomic status (SES) have not changed over the past decades in The Netherlands. To create a culture of health and an environment in which all people can flourish, a shift in focus is needed from disease management towards health promotion. The Bolk model for Positive Health and Living Environment was used as a tool to guide this shift. This study aimed to describe how this model was used and perceived by stakeholders in a case study on an integrated health promotion approach for residents with low SES. Methods. An instrumental case study was undertaken in Venserpolder, a neighborhood in Amsterdam South East of approximately 8500 residents. A participatory action approach was used that allowed continuous interaction between the residents, health care professionals, researchers, and other stakeholders. The Bolk model is a tool, based on the conceptual framework of positive health, that was developed to guide health promotion practice. Its use in the case study was evaluated by means of semistructured interviews with stakeholders, using qualitative directed content analyses. Results. The Bolk model was found to be a useful tool to identify and map the needs and strengths of residents with low SES. The model facilitated the development and implementation of eight health promotion pilots by transforming the needs and strengths of residents into concrete actions carried out by responsible actors in the neighborhood. Although the Bolk model seemed to be accepted by all stakeholders, the shift towards positive health thinking appeared to be more embodied by local professionals than by residents. Adjustments were proposed to enhance the applicability of the model in a multicultural setting, to increase its cultural sensitivity and to use language more familiar to residents. Conclusions. The Bolk model for positive health and living environment seems to be promising in the guidance of health promotion practices in Amsterdam South East. Further research and development are needed to improve its cultural sensitivity and to investigate its applicability in a broader range of public health settings.

## 1. Introduction

Growing up and living in socioeconomically deprived neighborhoods has been shown to have a negative impact on health. Residents living in those deprived areas have a lower life expectancy, number of years in good health, general health status, and higher morbidity [1]. Children of families with a low socioeconomic status (SES) are at higher risk to have a low birth weight, and the experience of childhood injury and abuse is twice as high compared to children growing up in families with higher SES backgrounds [2]. Despite efforts, health disparities between people with high and low SES have not changed over the past decades in most Western countries [3,4].

Amsterdam South East (SE), a relatively large district in Amsterdam with over 80,000 inhabitants, is characterized by a higher ethnic diversity, a higher percentage of residents with low SES, more single-parent families, and a higher unemployment rate than other areas of Amsterdam [5]. Amsterdam health monitoring data reported in 2012 were used at the time of the start of the project in 2016. These data show that adults 19 years of age and older smoke more, exercise less, eat less vegetables and fruits, and are more lonely and deprived of social contacts compared to people in other areas of Amsterdam [5]. A substantial amount of these adults have limited health literacy and feel as if they have little control over their lives [5]. Adolescents growing up in Amsterdam SE eat less healthy foods, exercise less, think more often about committing suicide, and feel less safe than adolescents in other parts of Amsterdam [6]. Consequently, the prevalence of obesity and other chronic diseases is higher in Amsterdam SE [5]. Given that inhabitants in Amsterdam SE experience suboptimal conditions with respect to social, physical, mental, and environmental factors, there is likely a strong link between these factors and their generally lower health status compared to others [5,6].

Until now, health promotion strategies have not been able to improve the general health status and health disparity of residents in Amsterdam SE [7,8,9]. A 2015 report demonstrated that obesity rates in adolescents from Amsterdam SE were as high as in 2009, and the sense of unease and safety among residents had increased [10]. The lack of positive health effects of community-based interventions may be explained by the fact that initiatives are often unfocused, unsubstantial, and short-term [8]. It is also difficult to reach out to residents with low SES. Since people with a variety of ethnic backgrounds live in the area, language is often a barrier for people to understand information about their health and to adequately explain how they feel to health officials [5]. This may hamper their access to professional health care. Another explanation for the lack of effect of community-based interventions is that those that need it the most are most socially isolated and may not easily find their way to social activities or other events organized in the neighborhood. Strategies on how to develop health literacy of residents with low SES and how to improve their health and lives remains an enormous challenge for all parties involved in the Amsterdam SE district.

A multidimensional, holistic, health promotion approach is advocated by the WHO to address health issues in low-SES neighborhoods and to decrease health disparities [11,12,13]. Such a multidimensional approach should include the physical and social environment, which have been shown to have a distinct positive impact on health in low-SES populations [14,15]. Factors such as improved social conditions at a district level, housing quality, sufficient playgrounds, access to public transport, biking opportunities, meeting places, and a safe and clean neighborhood can improve health of low-SES populations [8]. Research indicates that access to green spaces may also promote health and protect people from the negative health consequences of living in poverty [16]. However, a study assessing the effects of green space interventions in 36 Dutch areas on general health in low-SES neighborhoods did not find short-term effects on physical activity and general health [9].

Eliminating health disparities in low-SES populations thus requires a shift in focus from disease management towards health promotion, with the aim to develop a culture of health and an environment which stimulates healthy behaviors [11]. Over the last years, Positive Health has been introduced in the Dutch public health sector. Positive Health is based on the dynamic health concept as developed by Huber et al. [17]: health as “The ability to adapt and to self-manage, in the face of social, physical and emotional challenges”. It has been operationalized as a conceptual framework with six health dimensions: bodily functions, mental functions and perception, spiritual/existential dimension, quality of life, social and societal participation, and daily functioning [18]. The public health sector has embraced Positive Health as a promising conceptual framework to broaden health promotion programs to create a stronger position for citizens and inform research programs [19,20]. Although Positive Health has gained increasing interest among public health actors, it remains to be investigated how this conceptual framework can support health promotion in practice [21]. Therefore, the initiative was taken in 2016 to explore the use of the Bolk model for Positive Health and Living Environment (hereafter Bolk model) in the development and implementation of an integrated health promotion approach by means of a case study in Amsterdam South East. The Bolk model is a tool, based on the conceptual framework of Positive Health, that was developed to guide health promotion practice. It was hypothesized that this model would inspire and guide the large variety of health, prevention, and welfare actors in the South East district to co-create a health-promoting environment in collaboration with residents.

The aims of this study were as follows:(i)To describe the role and use of the Bolk model in the development and implementation of an integrated health promotion approach in Amsterdam SE.(ii)To describe the experiences and perceptions of residents and other stakeholders in the community where this model has been implemented.(iii)To explore whether a cultural transformation had occurred from “disease management thinking” towards “Positive Health thinking” and towards better collaboration among stakeholders in Amsterdam SE.

## 2. Methods

### 2.1. The Bolk Model for Positive Health and Living Environment

The Bolk model was developed by the first and second author, who are employed at the Louis Bolk Institute, as a tool for guiding health promotion practice. Inspired by the health field concept developed by Lalonde [22] and the ecological systems theory by Bronfenbrenner (Bronfenbrenner, 1979), the physical and social environment were introduced into the model and structured as surrounding spheres with increasing distance from the individual. Central to the Bolk model are the six domains of the conceptual framework of Positive Health: bodily functions, mental functions and perception, spiritual/existential dimension, quality of life, social and societal participation, and daily functioning [18]. The framework of Positive Health can be used by health care professionals to start a discussion with an individual client or patient about health [17]. When it concerns the Positive Health of a group of people living in a neighborhood setting, the model is used at a community level where the interaction with the physical and social environment is taken into account. The spheres can be chosen depending on the size and the specific characteristics of the area that is subject to an integrated health approach. The Bolk model that was used for the Amsterdam South East case study (see Figure 1) is presented with three surrounding geographical spheres: house, neighborhood, and city district. These spheres were considered appropriate for the case study, since they dealt with individual residents (house level) within a single district (Venserpolder: neighborhood level), and with health care organizations and the city government (city district level). Depending on the specific setting, spheres can be added to this dynamic model, such as street level, province, or country level.

### 2.2. Case Study—Setting

The case study took place in Venserpolder, part of the Bijlmer-Centrum neighborhood in Amsterdam South East with approximately 8500 residents. General health is experienced as good or very good by 64% of residents in Venserpolder, compared with 74% in Amsterdam collectively (±830,000 residents) and 75% in The Netherlands (±17 million residents). Of the population in Venserpolder, 13% are at a high risk for anxiety or depression disorders, compared to 9% in Amsterdam, collectively, and 7% in The Netherlands. Loneliness is reported by 21% of Venserpolder residents, compared to 13% in Amsterdam and 10% in The Netherlands [23].

### 2.3. Case Study—Design

An instrumental case study design uses a “case” to provide insight in the “issue of interest” and to facilitate the understanding of “something else” [24]. In the present study, the “case” was the development and implementation of an integrated health promotion approach, the “issue of interest” was the use of the Bolk model, and the “something else” was the cultural transformation towards Positive Health in Amsterdam SE. The case study had a participatory action research approach [25,26]. This approach was deemed most suitable since it allowed continuous interaction between practice (local organization of the integrated health promotion approach) and research (collecting and analyzing data). The case study consisted of four phases covering a total period of three years: phase (1) mapping needs and strengths of residents (5 months), phase (2) development of the integrated health promotion approach (10 months), phase (3) implementation and evaluation (12 months), and phase (4) sustainment of the integrated health promotion approach (9 months) (Figure 2).

### 2.4. Case Study—Participatory Approach

The goal of this study was to engage all relevant stakeholders in the four phases of the case study, in which each contributed on equal terms to the process of knowledge generation. The idea for the case study originated from a primary health care director working in Amsterdam SE. It was motivated by an intrinsic drive to empower residents in the area, as well as by the observation that the poor health status of the residents was partly caused by a lack of systematic communication between residents, formal, and informal care. Subsequently, contact was established with researchers of the Louis Bolk Institute to discuss whether the Bolk model could be a useful tool to co-create a health-promoting environment in Amsterdam SE with residents and stakeholders from formal and informal care. The study aims and plans were developed by two researchers from the Louis Bolk Institute, the health care director, and two local organizers who were recruited on the basis that they had the necessary contacts and trust among the residents and other stakeholders in Amsterdam SE. The development and implementation of the integrated health promotion approach was planned as a bottom-up process, initiated and driven by the residents in Amsterdam SE.

A project core team was established upon the start of the study. It consisted of three researchers including a project leader, the health care director, and two local organizers (Figure 3). The team was complemented with experts from local health care organizations, city district government, and social organizations. Prior to the start of the study, a local resident team was established that consisted of 5–7 residents (Figure 3). Residents were recruited by the local organizers via personal contacts and social media. Most residents in the local resident team were already actively involved in various volunteer and/or paid positions related to the health and wellbeing of residents in Amsterdam SE. Participating residents had interest in becoming involved in this project to further contribute to a better neighborhood. Furthermore, an advisory board was established that consisted of four experts in the field of public health. Additionally, a large number of local professionals and residents were attracted via personal contacts, as well as social media, to participate in the case study.

At a start-up meeting, the different roles and tasks of stakeholders in the case study were discussed and agreed upon. The core team was given the task to manage the (logistic) continuation of the project, to support the ongoing activities, to involve stakeholders, and to collect the research data in each phase of the study. The local resident team was given the task to advise the core project team, participate in the development and implementation of the integrated health promotion approach, to reach out to and engage other residents, contribute to research data collection, and take ownership of project activities. The advisory board was given the task to guide the project team in each phase of the case study and was consulted yearly.

#### 2.4.1. Phase 1: Mapping Needs and Strengths of Residents in the Amsterdam South East District

The aim of phase 1 was to identify the target population and their key needs and strengths by means of interviews, focus group discussions, and organization of two neighborhood tours (only the interviews are presented here). A qualitative research design with semistructured in-depth interviews was conducted by two researchers with 26 residents (13 male, 13 female), age between 21 and 69 years old (average 50 years old). A convenience sample was recruited through contacts of the local expert team, social media, and face-to-face communication in community centers and at neighborhood activities. Interviews were conducted in community centers in the neighborhood. Interviews were recorded and lasted between 17 and 159 min (average 44 min). The topics of the interviews were related to the six domains of Positive Health (see above) [18], and extended with questions about the physical environment, services, and social cohesion in Venserpolder (see Appendix A for the interview guide). The interviews were transcribed and a deductive content analysis [27] was performed by one researcher using Nvivo (1.0) software for qualitative data analysis in order to code and thematically group the strengths and needs of residents. The outcome of the content analyses was presented to the participants during a meeting in a neighborhood community center. During this meeting, the strengths and needs of residents were mapped by means of the Bolk model (Figure 1).

#### 2.4.2. Phase 2: Development of the Integrated Health Promotion Approach

The aim in phase 2 was to develop a bottom-up health promotion approach for residents in Venserpolder, with a focus on reaching residents with low SES, that aligned with their needs and strengths as determined in phase 1. The health promotion approach was guided by the Bolk model and designed by residents themselves by means of health promotion pilots (HPP). A health promotion pilot (HPP) is defined as a pilot project with local activities that aims to support and benefit the health of residents. Furthermore, it built on existing initiatives in the district, it aimed to reduce the distance between residents and health care professionals and organizations, and focused on an integrated approach, defined as strategic and operational partnerships and measures to promote Positive Health, healthy behavior, and health literacy, as well as improving the living environment that affects this behavior.

#### 2.4.3. Phase 3: Implementation and Evaluation of the Integrated Health Promotion Approach

The aim in phase 3 was to evaluate the experiences and perceptions of residents and other stakeholders in the community by working with the Bolk model, and to explore the process of cultural transformation from “disease management thinking” towards “Positive Health thinking” and towards better collaboration among stakeholders in Amsterdam SE. One researcher collected the data through observations of three project team meetings, minutes from project team meetings (8), core team meetings (2), local expert team meetings (2), advisory board meetings (2), HPP development and evaluation meetings (10), a cultural sensitivity workshop (1), stakeholder meetings (2), from the project communication plan (1), and by means of 15 semistructured interviews (see Appendix A for interview guide). A sample of participants for the interviews was recruited from project team members, local professionals, and residents that actively participated in the project [28]. Interviews were transcribed, coded (Atlas.ti software), and directed content analysis was applied [29]. Directed content analysis was based on the understanding, acceptance, and integration of Positive Health in the thoughts and behavior of residents and other stakeholders in the community [30,31,32,33]. A cultural transformation towards Positive Health thinking in Venserpolder was analyzed according to the Kotter change model. This model was developed by the Harvard Business Professor John Kotter with the purpose to guide leaders and organizations through a process of transformation. [34]. This model was mainly chosen because it can be operationalized in complex systems such as the current case study, in which all factors are interrelated. Furthermore, it was deemed more feasible to use a model which employs a chronological order in the process. The Kotter change model describes an eight-step process leading to change which was applied in this study as follows: (1) Urgency was explored through the identification of the motivating factors for change. (2) The guiding coalition was explored through composition and collaboration of stakeholders. (3) The vision was analyzed for its consistency and for being concise and desirable. (4) The communication of the vision was explored according to its repetition and two-way communication. (5) The barriers were identified in terms of structural barriers and enabling factors. (6) The last three steps in the Kotter change model (short-term wins, producing more change and sustainability) were combined into an overall evaluation of the sustainability of Positive Health thinking.

#### 2.4.4. Phase 4: Sustainment of the Integrated Health Promotion Approach

The aim in phase 4 was to achieve sustainment of the integrated health promotion and Positive Health approach, after the project was finished [35]. First, organizational and contextual characteristics related to health promotion activities in Amsterdam SE were identified. Secondly, relationships and collaborations were established with formal and informal health actors, the city district, and other stakeholders in the SE district and advocated to support their self-organizing ability and synergy between the various organizations. Subsequently, activities, information, and documentation as generated during the project were made available to all relevant stakeholders and, where possible, Positive Health thinking was incorporated within policies, missions, visions, and research activities of organizations and relevant stakeholders in the Amsterdam South East district.

### 2.5. Ethical Consideration

The case study was approved by the management board of the Louis Bolk Institute. They decided that this case study did not involve experiments with patients or study subjects according to the Dutch Medical Research Involving Human Subjects Act (WMO), and therefore was exempt from further medical ethical review. For the in-depth interviews, participants were informed about the purpose of the interview, that participation was voluntary, and confidentiality was guaranteed in the way that presentation of findings was anonymous. Participants were also informed that they, whenever they wished, could abort participation without explaining cause, and that it would not affect their participation in the case study. All participants provided verbal informed consent for storing and analyzing interview and demographic data.

## 3. Results

### 3.1. The Role and Use of the Bolk Model in the Development and Implementation of an Integrated Health Promotion Approach

With regard to the first aim of the study, the following subchapters describe the role and use of the Bolk model in the development and implementation of an integrated health promotion approach in Amsterdam SE.

#### 3.1.1. Identifying Needs and Strengths of Residents of Residents by Means of the Bolk Model (Phase 1)

First, interviews to identify the key needs and strengths of residents with low SES were guided by the six domains of Positive Health in the Bolk model (Figure 1; Appendix A). Content analysis of the interviews identified five major themes related to the needs and strengths of residents in the district Venserpolder. These were appearance of the neighborhood, social cohesion, help and care, knowledge and self-development, and organizations and institutions in the neighborhood. Each theme, with subsequent needs and/or strengths and illustrative excerpts from interviewed residents, is further described below.

##### Appearance of the Neighborhood

Key strengths, according to residents that were categorized under the theme appearance of the neighborhood, were safe neighborhood and colorful neighborhood. Residents recognized that safety could have an influence on health. Most interviewees reported feeling safe in the neighborhood; however, there were signals that people would not dare to talk about unsafe feelings. Residents mentioned that having people out in the streets gave them a safer feeling, as well as knowing the local police officers.

Residents mentioned that having parks, green areas, gardens, and art works motivates people to go outside. Most houses have recently been painted with various colors and a colorful piece of art was installed in the shopping area.


*Resident 1: “If you look outside and the weather is dark, you get dark yourself. But if the doors are closed and the windows are closed and the flowers and trees are nice and green, that makes me happy, it makes me smile. And then you go outside much happier.”*


However, some residents were not satisfied and became depressed from the appearance of the neighborhood. Key needs that were identified under this theme were less littering, greener neighborhood, and better parking options. Littering was reported as one of the major complaints giving rise to much irritation. Much dissatisfaction amongst residents was caused by lack of parking options for local residents and the use of parking spaces in the neighborhood by outsiders during events in neighborhoods close by.

##### Social Cohesion

Key strengths, according to residents that were categorized under the theme social cohesion, were contact with neighbors and contact with loved ones in home countries. Residents especially valued knowing their neighbors, greeting each other, and meeting each other at events, in community centers, in the streets, and in the shopping area. Neighbors replaced the loss of contact with family living in other countries.


*Resident 2: “I think people meet each other in community centers and there is a street culture here. Many people chat with each other on the street. At Multibron [read: a community center] we have a lot of accessible activities such as a chat hour.”*



*Resident 3: “I always make contact with everyone. When I go out in the street, I say: good morning, good day.”*


Safety in the neighborhood was frequently mentioned as a requirement for social interaction. It was not always perceived as easy to contact neighbors about unacceptable behavior.

Key needs under this theme were less loneliness, feeling at home, more opportunities for participating in activities, more places to meet, and more benches. Since the street is an important meeting place, placing more benches was seen as an option for stimulating social cohesion and decreasing loneliness. A sense of feeling at home could be increased by improving a feeling of community.

##### Help and Care

Support from neighbors and family and the role of key residents were regarded as the main strengths related to help and care in the neighborhood. An active network of mothers that provides support, such as taking care of each other’s kids and looking out for them when they played outside, was mentioned.


*Resident 4: “It is so nice that there are parents that are really “the help parents” in the school, they are like the mother of your child at that moment. If there is something [read: that they know about your child], and they know you or they know another mother, then it reaches you, (…) because you don’t always hear everything from the teacher.”*


Furthermore, it was deemed very important that there is a group of key residents in the neighborhood with a good overview of what is happening in the neighborhood, which offers support to many people.

Needs to improve help and care were identified as being able to feel comfortable asking for help, knowing ways to find help, knowing ways to overcome the language barrier, and having spaces for safe walking. Several residents mentioned that a lot happens behind closed doors. More social control and more trust between neighbors were seen as solutions for people to obtain help more easily. People rarely ask neighbors for help when they are ill; they prefer to rely on family members. However, many people do not have family members, and have to deal with the situations alone. There is some fear for other cultures in the neighborhood, which makes some people feel unsafe outside. A large number of people in the street make some people feel less safe, while others feel safer.

##### Knowledge and Self-Development

The key strength mentioned by residents related to knowledge and self-development is religion. The needs in this area were balance in life, knowledge about healthy living, and less debt-related stress.


*Resident 4: “You should work with the parents. Health can only be improved if you can get parents to change their mind setting as awareness of what is in the food products, how do I prepare it, how much do I eat, how do I balance it throughout the day and how do I combine it with exercise and how do I communicate my limits to my children.”*


Residents mentioned that many people lack knowledge about healthy foods and do not understand food labels. Furthermore, there was a lot of stress related to financial problems which is not perceived by formal care organizations.

##### Organizations and Services in the Neighborhood

Sport clubs, public transport, and active community centers were valued as the main strengths related to organizations and services in the neighborhood. The neighborhood has good public transport with two metro stations, a nearby train station, and multiple bus lines, which was perceived as one of the nicest aspects of the neighborhood. There is a sports center just outside the neighborhood, there are a lot of group activities, and several fitness machines in the park and several children’s playgrounds. However, people felt that not many people are actively pursuing a sport. People valued the health care center in the neighborhood, even though communication between the center and residents could be improved. Residents generally did not easily make use of formal care because of fear and shame. Community centers were especially mentioned to play an instrumental role in the wellbeing of residents, since they were run by a group of dedicated residents with many contacts within the neighborhood.

Needs regarding organizations and services were better communication between health and welfare organizations, and availability of, and access to, healthy food. Not many people knew about all the low-cost activities that are organized in community centers. There is a shopping area in the neighborhood. Residents mentioned that availability of healthy products in the shopping area is limited. There is a snack bar, which attracts a lot of younger people from the schools nearby. It was also noted that communication could be improved between the three community centers in the neighborhood. Formal care organizations often were not aware of activities in community centers. Therefore, there was a great need to connect formal and informal care. *Resident 2: “Community centers and social work are too far removed from some people. It is too formal. People from Ghana or the Antilles live in a very informal society. They just want to chat to someone in the neighborhood.”*

#### 3.1.2. Mapping Needs and Strengths of Residents Using the Bolk Model (Phase 1)

Second, in a focus group meeting with the interviewed residents, all key needs and strengths were discussed and placed within the respective sphere and Positive Health domain of the Bolk model (Figure 4). This visual map shows the distribution of the topics over the six Positive Health domains where strengths are mainly found in the quality of life, and social and societal participation domains, while many of the needs are found in other domains such as the mental health, physical health, and daily functioning domains. The map also provides the insight that a large number of needs can actually be addressed and influenced directly by individuals and families or by actors in the neighborhood, and that there are much fewer needs that would need to be addressed through involvement of the city district (see Figure 4).

#### 3.1.3. Development of Health Promotion Pilots (HPPs) by Means of the Bolk Model (Phase 2)

Thirdly, the Bolk model was used to guide the development of HPPs. Based on the Bolk model map of needs and strengths (Figure 4), a meeting was arranged with all involved stakeholders to discuss which spheres in the model could be influenced by residents themselves, which support they needed from other stakeholders, and who would take the initiative and lead for HPPs. For health promotion actions within the inner circles of the model, this could be a resident. For interventions more in the outer spheres of the Bolk model, it was more likely that a professional, for example a district manager, would take the lead. A total of eight HPPs were developed (Table 1). Presentations of the ideas from the local expert team were filmed and presented to the professional organizations during a focus group session. Each of those movies clearly showed the passion of the involved resident for their own projects. Each professional organization then adopted one or more of those ideas that matched with its own interests. Project teams were formed for each of the pilots, which consisted of at least one resident from the local expert team and one representative from a professional organization. The project team further developed the pilots during and after the focus group session. The project team assessed the context for successful implementation of the HPPs and facilitated matchmaking between professional organizations and residents.

#### 3.1.4. Implementation of the HPPs by Means of the Bolk Model (Phase 3)

Fourthly, the Bolk model was used to guide the implementation of HPPs. The implementation of an example HPP—the coffee hour in the health care center—is visualized in Figure 5. For this HPP, the street level was considered relevant and was therefore added as an extra sphere compared to the Bolk model, as depicted in Figure 1. The numbers in Figure 5 indicate the phases in the development and implementation of the HPP. The red arrows indicate the development in time. The HPP started with a group of residents that were running a coffee hour in a local community center (Figure 5: step 1). This was a street-level activity mainly focused on getting together and enjoying social interaction. This group of residents, represented by one resident in the local expert team, expressed the desire to do more with the subject of health within this coffee hour. They desired more information about health-related topics, more support with health issues, and more activities to improve health.

The project team identified the “social prescribing” activity, an activity in which residents are referred by a general practitioner to a range of local, nonclinical services to support their health and wellbeing (Figure 5: step 2) [36]. This social prescribing was offered by MaDi, a professional organization supporting residents with financial, social, and personal problems. Professionals involved in social prescribing expressed how difficult it was for them to reach out to vulnerable residents. The activity was organized from the office of MaDi, which is located a few miles outside of Venserpolder and serves an entire district of the city. The project team connected professionals from social prescribing with residents running the coffee hour in a community center. During the process it became clear that the residents could benefit from expertise and support available from social prescribing, and that the coffee hour could be an excellent way for the professionals to get in touch with residents in Venserpolder.

The project team subsequently contacted the local health care center, which also expressed a desire to reach out to vulnerable residents more easily (Figure 5: step 3). Therefore, the health care center offered free use of a space for conducting the coffee hour and offered to provide health-related information through presentations and leaflets.

The combination of the coffee hour run by residents, residents’ desire for health-related information, social prescribing needing to reach residents, and the health care center desiring to inform residents resulted in a joint coffee hour at the health care center, run by residents, with a social worker present to support people (Figure 5: step 4). The 10–15 residents attending the coffee hour each week brought their contacts with other residents within the neighborhood to the attention of the social worker if support was needed. Residents also organized swimming and walking groups with participants of the coffee hour to improve health.

The implementation of the other HPPs followed a similar process and guidance by the Bolk model (see description of the other HPPs in Appendix A).

### 3.2. Experiences and Perceptions of Residents and Other Stakeholders in the Community of Working with the Bolk Model (Phase 3)

With regard to the second aim of the study, the experiences and perceptions of residents and other stakeholders in the community were explored related to working with the Bolk model through the understanding, acceptance, and integration of Positive Health in their thoughts and behavior.

#### 3.2.1. Understanding of Positive Health

Interviewed (health care) professionals in Venserpolder were able to describe the underlying values of Positive Health. Through interviews with the active residents of Venserpolder, it appeared that some residents were familiar with the six dimensions of Positive Health, while other residents showed little to no understanding of Positive Health.


*Resident 9: “I do not know, maybe I do not understand [PH], maybe others do. I think [it is] a little bit difficult.”*


#### 3.2.2. Acceptance of Positive Health

Regarding acceptance, all those interviewed agreed that Positive Health can contribute to a positive change in the neighborhood.


*Resident 14: “Yes, I think it can [PH to bring change in Venserpolder], it makes you more conscious of the choices you can make to be more positive or aware of your neighborhood or with yourself.”*


Positive Health thinking and application of the Bolk model helped to structure and broaden the project and it stimulated collaboration. However, several adjustments were suggested to enhance applicability: these included increase cultural sensitivity of the Bolk model and use of language more familiar to residents.

#### 3.2.3. Integration of Positive Health in Thinking and Behavior

In general, the interviews showed an integration of the Positive Health concept in such a way that the view on health of those interviewed was in line with the key elements of Positive Health. Some aspects of health had become more concrete for the residents that were interviewed, and therefore easier to apply in daily life. Some interviewees mentioned that Positive Health thinking was seeded in the neighborhood. However, none of those had experienced a dramatic and sustainable cultural change in the neighborhood yet. Moreover, most interviewees explained that the different HPPs were attributed to change, and that it was part of an ongoing positive trend of change in the neighborhood. Integration of Positive Health seemed to be more effective among professionals than among residents.


*Professional 2: “I have to say, the research, for me personally it was an eye opener. Health is not just health; it is also wellbeing.”*


### 3.3. The Process of Cultural Transformation from “Disease Management Thinking” towards “Positive Health Thinking” and towards Better Collaboration among Stakeholders in Amsterdam SE (Phase 3 and 4)

With regard to the third aim of the study, the process of cultural transformation towards Positive Health and better collaboration were explored according to the Kotter model of change [34].

#### 3.3.1. Urgency

Most professionals expressed an urgency to improve the health status of residents in the neighborhood and to decrease the fragmentation of care initiatives in the neighborhood.


*Professional 1: “People live seven years shorter, and the quality of healthy living is fourteen years shorter I believe. Well, you do not need more reasons I think.”*


#### 3.3.2. Guiding Coalition

All respondents agreed that the Bolk model for Positive Health increased (transdisciplinary) collaboration. The Bolk model was perceived to facilitate collaboration because it clarified the need for collaboration and was a well-known and important topic that stakeholders wanted to work with. The model increased awareness of everyone’s unique role in supporting the health of residents.


*Researcher 1: “Yes I think one of the powers [PH] is that it creates connections between different organizations and stakeholders. The people feel seen, that they all form part of the solution and that the resident is central.”*


Initially, the driving force of the project consisted of a core team with good understanding of Positive Health. The core team had a strong network with both informal and formal care providers in the area and was trusted by the residents. Furthermore, the core team possessed good communication skills, was able to contact and communicate with various citizen groups and possessed strategic thinking.

#### 3.3.3. Change Vision and Communicating the Change Vision

The underlying values of the Bolk model, i.e., a broader outlook on health, identifying intrinsic needs, and empowerment of residents, were seen in all phases of the case study. However, the specific dimensions/spheres of the Bolk model and the conceptual model for Positive Health behind the model were often mixed up by respondents. The vision of Positive Health appeared to be communicated implicitly through the pilots, but the need to further translate the concepts and ideas to the residents’ language and understanding was emphasized. The popularity of the Positive Health vision among professional stakeholders was very helpful in facilitating collaboration, as well as the bottom-up communication of the vision.


*Project coordinator 1: “The system world speaks a different language than the target group whom it concerns. The communication has to be target-bound, the information that is made public should be accessible for everyone.”*


#### 3.3.4. Barriers and Enabling Factors

A number of barriers for cultural transformation towards Positive Health thinking were identified. Different missions of organizations could be conflicting, hampering the possibilities for collaboration between organizations. Interviewed civil servants expressed that organizations can only provide extra time for the project if the vision is in line with their own mission. Furthermore, many civil servants mentioned the fragmentation of initiatives, the lack of overview of initiatives, and lack of overview of organizations active in the neighborhood as barriers for effective collaboration. Barriers for empowering the residents were mostly formed by difficulties in successfully reaching the target group and creating ownership among them. All project coordinators expressed a lack of budget to invest sufficient time in the project. An enabling factor was that respondents emphasized that the project succeeded in bringing ownership to the residents and considered this a key success. Residents learned new skills and received sufficient support from the project team.


*Resident 15: “… The communication they [the project team] have with residents [is specifically good about this project in comparison to other projects], to involve residents and that they can take initiative to organize things and create workshops, etc.”*


#### 3.3.5. Sustainability

An important factor for sustainability was found to be the direct inclusion of the Positive-Health-based integrated health approach into the city district government district plans for 2019, ensuring possible government funding for future activities related to Positive Health. Positive Health was popular amongst the health care professionals, which enhanced collaboration and bottom-up working with residents. Furthermore, two active residents from the neighborhood trained in working with Positive Health were elected to the city district government. One other resident exposed to the Positive Health approach, representing disabled people and also active in an HPP, joined a local committee that deals with the city district government regarding the organization of the physical environment. Sustainability was also fostered by a WhatsApp group amongst participating residents and organizations, which developed into a Positive Health network and was used for sharing ideas, recruiting participants, and motivating each other.

## 4. Discussion

The case study in Amsterdam SE demonstrated that the Bolk model for Positive Health and Living Environment was actively used as a tool for identifying and mapping of needs and strengths of residents in Amsterdam SE. It visually guided stakeholders in the implementation of health promotion activities in a deprived neighborhood with a focus on empowering residents and fostering social cohesion and collaboration. The Bolk model added a broader perspective and vision on health to the existing development process of health promotion interventions. Rather than focusing on only one aspect of health, such as physical health, as achieved by more physical exercise, healthier eating, or smoking cessation [37], it takes into account multiple determinants of health, including, for example, spirituality and the living environment. In this respect, the Bolk model aligns with socioecologic approaches [38] and community-based health promotion interventions [39]. However, addressing a large variety of health-related factors is complex and difficult in health promotion practice [39]. Results from the present study indicate that the Bolk model might be a promising tool to target multiple determinants of health on different levels simultaneously. In addition, the Bolk model seems to facilitate a cultural transformation in health care toward a broader perspective on health. Such a cultural transformation in health care can also be observed in the UK where social prescribing is growing in popularity [40]. Social prescribing is a primary care service in the UK that links patients with nonmedical needs to sources of support provided by the community and voluntary sector to help improve their health and wellbeing [41].

Questions were frequently raised in this case study about whether all stakeholders should use and embody the concept of Positive Health and Bolk model in a similar way, or if one could rely instead on the underlying values of Positive Health, such as having a broad view on health (including physical, mental, and social elements), resilience, and focusing on the strengths of residents. Similar discussions were reported to take place in the Positive Health project in the Noordelijke Maasvallei (northern valley of the Maas river), The Netherlands [42]. In the present case study, the Positive Health domains within the Bolk model appeared to be presented in a manner too difficult to understand for certain groups of residents and might benefit from reducing and simplifying textual elements. It appeared that the underlying values seemed to be more important than the six dimensions of Positive Health in the current case study. Therefore, clarification of these values may facilitate alignment between stakeholders and may ease the communication to (new) stakeholders.

The results of this study should be interpreted in line with its strengths and limitations. A main strength of this study was that the Bolk model was evaluated in a relevant instrumental case setting, in which a variety of stakeholders were involved and in which the needs and strengths of residents played a crucial role. Furthermore, the case study showed the usefulness of the integrative nature of the Bolk model for mapping the needs and strengths of stakeholders and developing an integrative neighborhood approach. The multidimensional health concept, within the physical and social environment, facilitated bringing health care professionals and residents together and increased the focus on community assets. The Bolk model thus extended the Positive Health concept for individual use to a practical tool which can be used collectively for public health promotion. A major limitation was that the value and usefulness of the Bolk model in health promotion practice was not measured in a quantitative way in the case study. Attempts to involve residents in setting up quantitative monitoring and evaluations indicated a low priority of residents for research, as well as difficulties in getting residents to understand the importance of the research process. Furthermore, it frequently felt too intrusive to request data from vulnerable residents.

When the case study came to an end, it was clear that much still needed to be carried out in the neighborhood. Despite involving co-researchers from the neighborhood, training residents, focusing on the strengths of residents, involvement of health care professionals, and having support from the city district government, it is very challenging to create sustainable changes in a low-SES multicultural neighborhood. Health data obtained from Amsterdam SE district in 2020 [43] showed no improvement in health outcomes compared to the 2014 data [5]. Overweight had increased from 48% to 52% of the Amsterdam SE population, severe loneliness increased from 15% to 18% over the years, and low control over their lives increased from 14% to 16% [5,43]. Health disparity between Amsterdam SE and the rest of Amsterdam has thus not been reduced in the last decade. However, the fact that health indicators do not seem to have changed in Amsterdam SE can be attributed to a large range of factors and events. Changes in obesity rates and control over life may be determinants of health that cannot simply be solved by implementing an integrated health promotion (such as in the current case study) over a period of three years. Follow-up studies are therefore warranted to quantitively explore the relationship between other variables and related health indicators, such as loneliness, and to spread the integrated health promotion approach from Venserpolder to the whole district of Amsterdam SE.

Throughout the case study, several suggestions were given to further optimize the Bolk model for use in health promotion practices in (multicultural) neighborhoods. Most importantly, respondents mentioned the lack of cultural sensitivity of the Positive Health concept. It has previously been criticized for its individualistic perspective [44], which is characteristic for the Western culture and society in which it was developed [18]. However, in multicultural neighborhoods such as Amsterdam Venserpolder, considerable attention is given to the collective perspective: “I am only healthy when my family, neighborhood, and friends are healthy”. Further research is therefore recommended to specifically develop and validate working tools as part of the Bolk model that are more suitable and applicable to the cultural differences within a community. The Hofstede model [45], with its six dimensions of national cultures, could be an appropriate model to guide such development of working tools. Future research should also involve the implementation and evaluation of the Bolk model in other community approaches, such as in asset-based community development (Agdal et al., 2019). Furthermore, it is recommended to investigate whether the Bolk model is useful in working with community resilience, which is characterized by a demand-driven, people-centered approach [12,46,47]. The reported focus on self-management of a local resident team, partaking in the city district government and other community organizations, as well as the exposure of residents to the HPPs in the present case study, may have improved the community resilience of Venserpolder. However, quantitative monitoring, as well as long-term studies, are necessary to investigate the contribution of the Bolk model to community resilience and health-related quality of life of vulnerable populations.

## 5. Conclusions

The Bolk model for Positive Health and Living Environment seems to be a promising tool for health promotion practice in socioeconomically deprived neighborhoods. It is a comprehensive but practical tool that is of support to identify options for improvements in neighborhoods and to systematically develop solutions that can be used to support local stakeholders. Further development and research on the Positive Health concept and the Bolk model are needed to optimize their use within a multicultural setting and to investigate their impact towards guiding changes in health-related quality of life in the long term.

## Figures and Tables

**Figure 1 ijerph-19-02478-f001:**
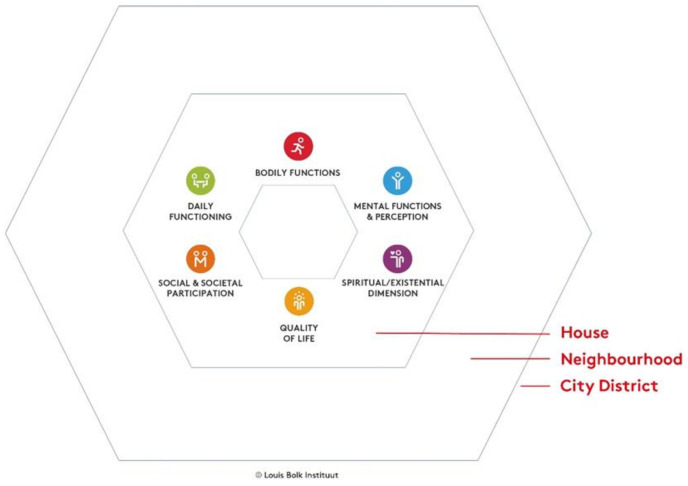
Bolk model for Positive Health and Living Environment: framework.

**Figure 2 ijerph-19-02478-f002:**
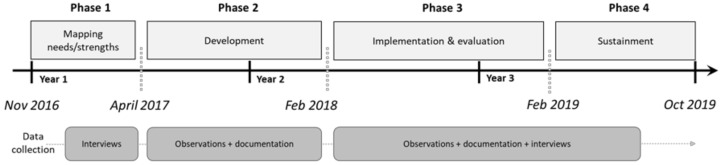
Schematic overview of the timeline, phases, and data collection in the case study.

**Figure 3 ijerph-19-02478-f003:**
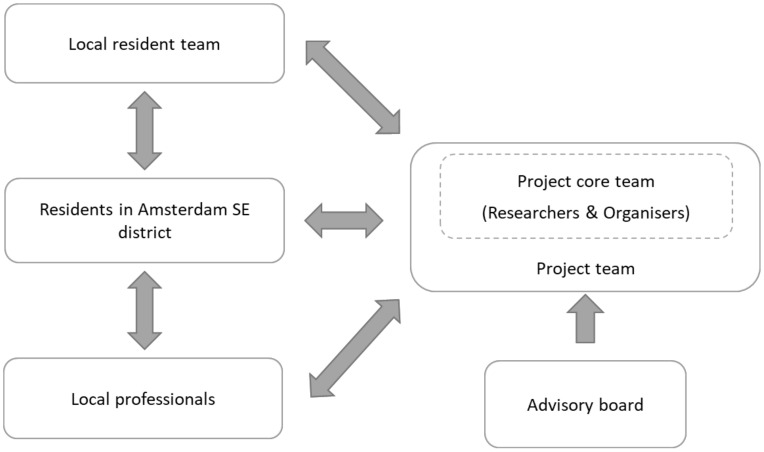
Schematic overview of the participatory approach in the case study.

**Figure 4 ijerph-19-02478-f004:**
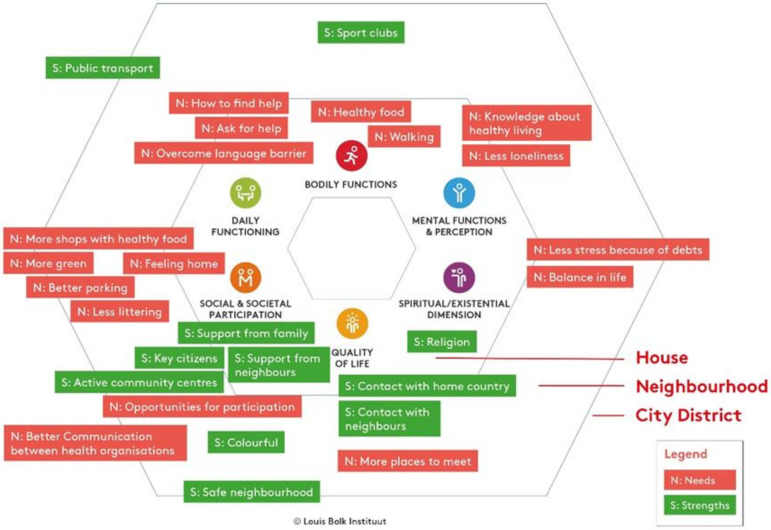
Mapping needs and strengths of residents in the Bolk model.

**Figure 5 ijerph-19-02478-f005:**
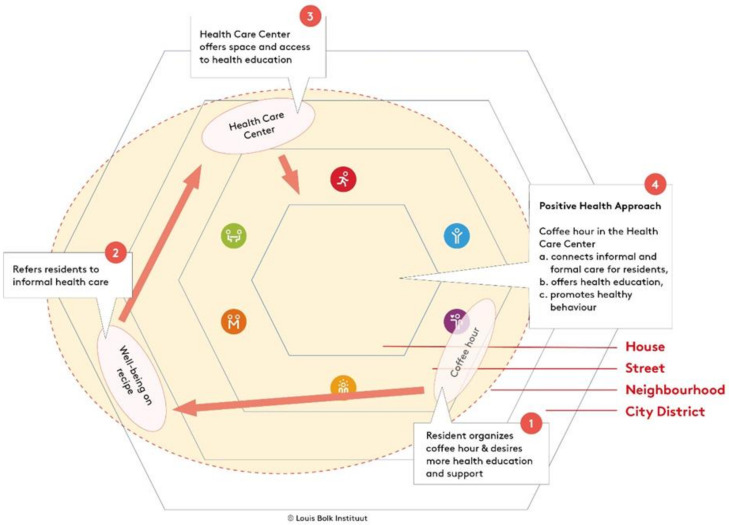
Implementation of a health promotion pilot. Note: An example illustration of how the Bolk model guided implementation of the HPP: coffee hour in the health care center. The four-step process is described in more detail under Section 3.1.4.

**Table 1 ijerph-19-02478-t001:** Overview of the health promotion pilots in the case study.

Pilot	Expected Result	Target Group
1. Coffee hour in health care center	Informal talk about health and social support, improved access of health care professionals to residents	Residents Venserpolder
2. Digital connecting	Digital map of informal care options and social activities available in the library	Residents Amsterdam South East
3. Healthy shopping area	More healthy products in shops in the local shopping area	Residents Venserpolder
4. Green and health workshops	Vegetable gardening workshops,	Women Venserpolder
5. Healthy eating for kids	Teaching children food preparation and healthy eating	Children Venserpolder
6. Man power	More activities for men	Men Venserpolder
7. Accessibility	Better access to streets and buildings for people with special needs	Residents with special needs
8. Thematic health events	Knowledge sharing about health and personal development	Residents Venserpolder

## Data Availability

Detailed reports concerning the data of the process evaluation and the mapping of needs and strengths can be requested from the corresponding author.

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
