# Peer review of "The Use of the Bolk Model for Positive Health and Living Environment in the Development of an Integrated Health Promotion Approach: A Case Study in a Socioeconomically Deprived Neighborhood in The Netherlands"

_ijerph, 2022, doi:10.3390/ijerph19042478_

Round 1
Reviewer 1 Report
This is an interesting case study of a multi-phase intervention. The participatory nature of the research is very compelling. Overall, it is well written, but the language and grammar are awkward in places (some examples below) and would benefit from editing.
- Whenever possible, move from passive to active voice. This is a particularly valuable approach when describing participatory research. Examples of passive language that could be improved by specifying who led and was involved in activities:
Line 115: ‘Therefore, the initiative was taken in 2016 to investigate the use of the conceptual framework of Positive Health in the development and implementation of an integrated health promotion approach by means of a case study in Amsterdam South-East. It was hypothesized that this conceptual framework would inspire and guide the large variety of health-, prevention- and welfare-actors in the South-East district, as to co-create a health-promoting environment in a broad sense in collaboration with residents.’
150: ‘In the case study of Amsterdam South-East, the Bolk Model for Positive Health and Living Environment was first used as a tool to map the needs and strengths of residents in Venserpolder by means of interviews.
- Figure 5 is a bit confusing. There is a reference to blue arrows, but the arrows appear to be red. A description or figure that is more of an overview of how the pilots were implemented across sites might be more useful.
- The subheads in 3.3.2 aren't formatted well, making the section a bit confusing.
- Make sure terms and names are defined when introduced. For example:
Line 238 – Health Promotion Pilots
Line 439 - MaDI
- The introduction is very good overall. The only suggestion is to move from passive to active voice in describing the initiative. This would provide readers with information about who undertook this initiative (including the authors and any collaborators.
- In the methods section, (lines 156-157) could you explain how wanting more shops for healthy food aligns with ‘social & societal participation’. It is just not intuitive to me.
- Also in the methods section, please be specific about who was engaged in different parts of the process. For instance, lines 164-168: ‘Based on the final Bolk model map of needs and strengths, the model was further used to guide the development of interventions to promote Positive Health in the district. The spheres in the model were discussed in relation to what residents could achieve themselves, which help they needed from other stakeholders, and whom would take the initiative and lead for the suggested interventions.’ Who was involved in using the model to guide the development of interventions? Who was involved in the discussions?
- Some typos:
Line 130: ‘was developed by the first and second author which are employed at the Louis Bolk Institute.’ Replace which with who?
331: ‘Since the street is such an important meeting place, placing more benches were seen as an option for stimulating social cohesion, and decreasing loneliness.’ Replace were with was.
Line 454: residents' desire
485: contribute to?
509: take out 'also'?
598-601 ‘Authors should discuss the results and how they can be interpreted from the perspective of previous studies and of the working hypotheses. The findings and their implications should be discussed in the broadest context possible. Future research directions may also be highlighted. Does this belong in the text?’ Looks like notes.
- Examples of places to rewrite for clarity:
Line 62: A substantial amount of these adults have limited health literacy and feel like having little control over their lives
470: The evaluation of the use of the Bolk model in the implementation and sustainment of the integrated health promotion approach in Venserpolder concerned the evaluation whether a shift from ‘disease management thinking’ towards ‘Positive Health thinking’ has occurred among the stakeholders in Venserpolder.
Sections 3.32 and 3.4.
Reviewer 2 Report
The study provides a detailed description of the development of an integrated health promotion approach for residents with low SES in the Netherlands guided by the Bolk Model for Positive Health and Living Environment. However, there is some information that I found was missing that could strengthen the paper further. The manuscript requires some changes as indicated below.
Methods:
It is questionable whether a case study is appropriate as a research method. This study relied mainly on interviews with stakeholders who participated in the project, so I thought that it was a qualitative study as formative research. The key to case studies is to have cases with boundaries appropriate to the research problem. If an author intends to conduct a case study, the unit of analysis and the case to be studied should be clearly defined. In addition, theories, propositions, and research questions should be presented so that the results of case studies can be generalized. It is recommended that other data sources for case studies need to be included to cross-validate the cases and deepen understanding of the case.
The authors mentioned that the PAR design was used in the research design, but the design description including principles such as establishing an equal partnership between researchers and participants is lacking. The Resident's participation in the intervention project development process recruited by the convenient sampling method is not considered PAR.
Results:
The presentation is way too long, major points made are not easily detectable by the reader. But the main objection here is, that the analysis itself seems not structured or analyzed enough. To my understanding, in content analysis, the data should have
been analyzed and structured in more depth, therefore results are not clear enough at this stage. Especially, phase 2~4 should be rewritten.
In Figure 4, there are three categories such as house, neighborhood, and city district. However, in Figure 5, there are four categories including streets. The authors should clearly explain the differences between the two figures. In addition, the figure title should be revised.
Discussion:
It is difficult to find an approach that the author interprets or analyzes because the phenomenon is just described. If there are new aspects, those should be highlighted at the beginning of the discussion.
The fact that regional health indicators did not change despite innovative approaches must also be taken into account for several other variables. For example, changes in obesity rates and control over life may be health issues and determinants of health cannot be solved by implementing this project for only several years. Rather than pointing it out as a limitation of the project, I would like to suggest a follow-up study on the relationship between other variables and related indicators, or even provide a starting point for a plan to spread this approach throughout the region.
Minor points:
Even though the original text was written in Dutch, an English translation should be added so that readers can understand what the reference is.
Please check typos (e.g., focussing -> focusing)
Overall: This paper does not appear to be in a stage to be published at this moment. After major revision and improvement, it may be reconsidered.
Round 2
Reviewer 2 Report
Thank you for thoroughly addressing the points raised. Please see attached documents with a few detailed responses to your responses to my previous comments. After minor revisions, I feel that the paper will make a contribution to the Journal.
